# Silicon Supply Improves Nodulation and Dinitrogen Fixation and Promotes Growth in *Trifolium incarnatum* Subjected to a Long-Term Sulfur Deprivation

**DOI:** 10.3390/plants12122248

**Published:** 2023-06-08

**Authors:** Raphaël Coquerel, Mustapha Arkoun, Quentin Dupas, Fanny Leroy, Philippe Laîné, Philippe Etienne

**Affiliations:** 1Unicaen, INRAE, UMR 950 EVA, SF Normandie Végétal (FED4277), Normandie Université, 14000 Caen, France; raphael.coquerel@unicaen.fr (R.C.); quentin.dupas@unicaen.fr (Q.D.); philippe.laine@unicaen.fr (P.L.); 2Laboratoire de Nutrition Végétale, Agro Innovation International-TIMAC AGRO, 35400 Saint-Malo, France; mustapha.arkoun@roullier.com; 3Plateau Technique d’Isotopie de Normandie (PLATIN’), Unité de Services EMERODE, Normandie Université, 14000 Caen, France; fanny.leroy@unicaen.fr

**Keywords:** N_2_ fixation, nodules, S deficiency, silicon, ^15^N labeling, *Trifolium incarnatum*

## Abstract

In many crops species, sulfur (S) deprivation negatively affects growth, seed yield quality and plant health. Furthermore, silicon (Si) is known to alleviate many nutritional stresses but the effects of Si supply on plants subjected to S deficiency remain unclear and poorly documented. The objective of this study was to evaluate whether Si supply would alleviate the negative effects of S deprivation on root nodulation and atmospheric dinitrogen (N_2_) fixation capacity in *Trifolium incarnatum* subjected (or not) to long-term S deficiency. For this, plants were grown for 63 days in hydroponic conditions with (500 µM) or without S and supplied (1.7 mM) or not with Si. The effects of Si on growth, root nodulation and N_2_ fixation and nitrogenase abundance in nodules have been measured. The most important beneficial effect of Si was observed after 63 days. Indeed, at this harvest time, a Si supply increased growth, the nitrogenase abundance in nodules and N_2_ fixation in S-fed and S-deprived plants while a beneficial effect on the number and total biomass of nodules was only observed in S-deprived plants. This study shows clearly for the first time that a Si supply alleviates negative effects of S deprivation in *Trifolium incarnatum*.

## 1. Introduction

At the end of the 20th century, various policies were implemented to drastically decrease SO_2_ emissions into the atmosphere [1,2] in order to improve air quality and reduce the negative effects of these emissions on human health [3]. Consequently, this legislation has led to a strong decrease in sulfur (S) deposits on agricultural soils and has increased the occurrence of S deficiency [4], especially in plants requiring high S such as brassicas [5,6,7] and legumes [8,9,10,11]. A sulfur deficiency can lead to many metabolic perturbations [12,13] such as reduced atmospheric dinitrogen (N_2_) fixation [8,14] and nitrogen (N) uptake from soil [15] and it usually leads to sharp drops in the yield and quality of the harvested product [9,16]. Finally, due to the decrease in seed quality, sulfur deficiency has indirect negative impacts on the health of human and animal consumers [17]. Currently, use of S fertilizers at a dose of about 30 kg S ha^−1^ [18] is the most widely used strategy to support S deficient crops. However, in the global agroecological context of reducing inputs, it is urgent that other strategies be developed to alleviate the stress of plants experiencing S deficiency. Among them is consideration of supplementation with silicon (Si), whose beneficial effects on plants subjected to biotic [19,20] and abiotic [21] stresses has been widely described in the literature. For example, a recent review reported that Si mitigates the negative effect of abiotic stresses as early as seed germination [22] and several studies have reported that Si supply enables alleviation of plants experiencing drought [23] or heat stress [24]. In the same way, a Si supply enables better tolerance to salt stress by sustaining various biological process such as photosynthesis or by stimulating detoxification of reactive oxygen species (ROS) [25]. Moreover, it has been shown that Si supplementation is beneficial in plants subjected to nutrient deficiency, such as deficits in N, P and K, and to a lesser extent S [26,27,28]. In legumes, the beneficial effects of silicon have also been demonstrated in plants subject to a large panel of abiotic stresses. For example, in several legume species exposed to salt stress, Si supply leads to maintenance of atmospheric N_2_ fixation by increasing the number of nodules and their leghemoglobin concentration and nitrogenase activity [29,30]. Although Si has several beneficial effects under a variety of adverse conditions, there are currently no studies that have focused on the effects of Si application to legumes subjected to S-deficiency. Indeed, currently only three studies demonstrate beneficial effects of Si application in barley [31], rice [32] and rapeseed [28] subjected to S-deficiency. In this study, we focused on *Trifolium incarnatum* because previous studies have shown that this legume species is of agroecological interest especially when it cultivated in mixed crops. For example, in *Trifolium incarnatum*-*Brassica napus* intercrop, several works recently report shown that the transfer of nitrogenous compounds from clover to rapeseed improved the agronomic performance of this field crop [18,33,34].

The objectives of this study were to assess the effect of Si supply in *Trifolium incarnatum* subjected (or not) to a long-term S deficiency (63 days). For this, the effect of Si supply on the growth, nodulation (number, biomass and nitrogenase abundance of nodules) and atmospheric N_2_ fixation capacity was performed in S-deprived or S-fed plants and those supplied or not with Si (1.7 mM) over 21, 42 and 63 days. Finally, the beneficial effects of Si supply on growth, nodulation and N_2_ fixation observed in S-fed and mainly S-deprived plants have been discussed.

## 2. Results

### 2.1. Effect of Si Supply on Growth of Trifolium incarnatum Subjected or Not to S-Deprivation

At D21, whole plant biomasses were similar and reached about 5 g plant^−1^, regardless of the S and Si supplies. At day 42, irrespective of the Si supply (-Si or +Si), S-deprivation had no impact on whole plant biomasses (Figure 1). However, it is notable that the Si supply led to a significant increase in shoot biomass (around 1.5-fold) in both S-fed and S-deprived plants. In comparison to S-fed plants, at D63 a S-deprivation led to a significant decrease in total biomass when plants were cultivated without Si, which was as expected. In contrast, whichever S treatment was considered (+S or -S), Si supplementation resulted in an increase in total plant biomass (by 1.2- and 2-fold in +S+Si and -S+Si plants, respectively; Figure 1 and Appendix A). In S-fed plants (+S+Si), this effect is explained by an increase in shoot biomass alone. In S-deprived plants, this was partly the result of an increase in root biomass, but mainly due to a strong increase in shoot biomass. Interestingly, it could be highlighted that a Si supply offset the negative effect of S deprivation on plant growth because the total biomass of -S+Si plants was increased by around 13 g plant^−1^ compared to -S-Si plants, and reached a biomass similar to S-fed plants (24.75 g plant^−1^ ± 2.5 g) (Figure 1 and Appendix A). 

### 2.2. Determination of Total Sulfur (S) and Silicon (Si) Concentrations in Plants

As expected, the S concentration in roots and shoots in S-deprived plants were always lower than in those of S-fed plants (Figure 2A,B), whatever the harvest date (D21, D42 or D63). In general, Si supply had no effect on S concentrations in plants compartments except at D21 where the S concentrations in roots and shoots in -S+Si plants were slightly lower than those of -S-Si plants (Figure 2A,B). In the same manner, whichever S treatment was considered (-S or +S), the Si concentrations in roots and shoots of plants cultivated without Si (+S-Si or -S-Si) were always lower than those of plants supplied with Si (-S+Si or +S+Si). Moreover, in plants supplied with Si, it was evident that the Si concentrations in roots increased throughout the treatment (from 1.8 to 3.5 mg g^−1^ DW on average), while the Si concentration in shoots was always lower and remained stable (around 1.1 mg g^−1^ DW), regardless of the harvest date (Figure 2C,D). These data show that, as previously demonstrated in *Brassica napus* by [26], *Trifolium incarnatum* preferentially accumulates Si in roots. Finally, it was observed that Si concentrations were not affected by the presence or the lack of S in the nutrient solution (Figure 2C,D). 

### 2.3. Effect of Silicon Supply on Root Nodulation

In S-fed plants, regardless of the harvest date (D21, D42 or D63) or whether plants were treated with or without Si, the number of nodules per plant (approximately 7000 nodules plant^−1^; Figure 3A), the weight of a single nodule (Figure 3B), and the weight of all nodules per plant (Figure 3C) remained similar. Compared to +S-Si plants, in -S-Si plants, the number of nodules (around 4000 nodules plant^−1^), the weight of a single of nodule (Figure 3B) and the weight of all nodules by plant (Figure 3C) decreased for all treatment durations. These results showing a negative effect of S-deprivation on nodulation agree with previously reported work performed in *Trifolium repens,* another clover species [8]. In contrast, surprisingly, the provision of a Si supply to -S+Si plants alleviated the negative effect of S deprivation on the number of nodules per plant, which again was similar to the observations in S-fed plants at D21 and D42 (between 6000 and 7000 nodules plant^−1^) and even to a high increase at 63 days (15,000 vs. 7000 nodules plant^−1^ in -S+Si and +S-Si, respectively) (Figure 3A). In addition, while the addition of Si in these -S+Si plants did not affect the weights of individual nodules or even decreased their individual weights at D63, it nevertheless led to an increase in the total weight of the nodules per plant at D21 (around 2-fold), D42 (around 1.5-fold) and more especially at D63 (around 4-fold) compared to the -S-Si plants (Figure 3C). 

### 2.4. Effect of Silicon Supply on the Total Plant N Derived from Uptake and N_2_ Fixation 

Whatever the S treatment (+S or -S), at D21 the Si supply had no effect on either the amount of N taken up from nutrient solution or the amount of N derived from atmospheric N_2_ fixation (Figure 4A). In contrast, at D42, a Si supply led only to an increase in the N amount derived from atmospheric N_2_ fixation in plants cultivated with or without S. For example, at D42, in S-fed plants, the Si supply led to an increase in the amount of N_2_ derived from the atmosphere by around 1.8-fold and contributed to an increase in the total N amount from 296 ± 28 to 434 ± 23 mg plant^−1^ compared to +S-Si plants (Figure 4A). In addition, at D63, it can be pointed out that in plants grown without Si, S deprivation significantly reduced (by 1.5-fold) the total amount of nitrogen in plants from 644 ± 45 to 400 ± 41 mg plant^−1^. This decrease is explained by a decrease in the amount of nitrogen taken up from the nutrient solution and especially by a decrease in N_2_ fixation capacity, which dropped from 45 to 30% (Figure 4B). At this harvest time, irrespective of the S treatment (+S or -S), a Si supply increased the N_2_ fixation capacity, which resulted in an increase in the total N amount in plants. However, this Si effect was exacerbated in S-deprived plants. Indeed, in a spectacular way, supplying Si to S-deprived plants allowed a strong increase in both the amount of N fixed (around 2.7-fold) and the total N (around 1.8-fold), which reached values comparable to those observed in S-fed plants. Thus, by improving the N_2_ fixation of S-deprived plants (by 30–55%; Figure 4B), a Si supply mitigates the negative effect of S deprivation on N nutrition and thus maintains the total amount of N in -S plants at a level similar to that observed in +S plants (Figure 4A).

### 2.5. Effect of Silicon Supply on Abundance of Nitrogenase in Nodules 

At each harvest date and for each treatment, the concentration of nitrogenase in nodules (U.A. mg^−1^ FW) was determined using Western blot analysis (S2). From these data, at each harvest date, the total abundance sof nitrogenase in all nodules of one plant was assessed (see Materials and Method for details) and expressed relative to +S-Si plants (considered as controls) to study the effects of S and Si treatments on this parameter (Figure 5). At D21, a negative effect of S-deprivation on nitrogenase relative abundance was observed and Si supply had no effect on this parameter, irrespective of the S treatment. At 42 and D63, the negative effect of S-deprivation on nitrogenase relative abundance was observed only in plants cultivated without Si. Indeed, compared to -S-Si plants, the Si supply increased the nitrogenase relative abundance by 5-fold and 6-fold at D42 and D63, respectively. This increase is a consequence of the increase in nitrogen concentration in nodules (S2) and of the increase in the total nodule biomass (Figure 3C) in -S+Si plants compared to -S-Si plants. Interestingly, the nitrogenase relative abundance in -S+Si plants at D63 again became similar to the values observed in +S+Si plants, suggesting a new time period in which supplying Si alleviated the negative effect of S-deprivation. In S-fed plants, even though a Si supply also resulted in an increase (around 3-fold) in the nitrogenase relative abundance at D42 and D63 (Figure 5), this increase was only due to an increase in the nitrogenase concentration in nodules of +S+Si plants (S2) because the total biomass of nodules was similar in +S-Si and +S+Si plants (Figure 3C).

## 3. Discussion

Silicon is largely known to alleviate negative effects of abiotic stress in plants [26,28,35,36] and to improve plant health [30]. However, the effect of Si supply on S-deprived plants is currently poorly documented, as reported by [27,37]. The main objective of this study was to evaluate the effect of Si supply on the growth, nodulation and atmospheric N_2_ fixation capacity of *Trifolium incarnatum* subjected (or not) to long-term sulfur deprivation (63 days). This study demonstrated that the growth of *Trifolium incarnatum* was not affected by S deficiency during the 21 first days of treatment (Figure 1). This result agrees with previous work performed in *Brassica napus* cultivated in hydroponic conditions and subjected to the same duration of S-deprivation [28]. In contrast, when the S-deprivation was prolonged (D63), this study showed that a strong decrease in both shoot and total biomasses occurred in -S-Si plants compared to +S-Si plants (Figure 1). This result agrees with previous studies reporting that long-term S deficiency limits growth in several leguminous plant species [8,9]. A supply of Si to these S-deprived plants led to an increase in shoot biomass at D42, and more interestingly to an increase in root and shoot biomass at D63, which allowed -S+Si plants to reach a total biomass similar to and even higher than that of S-fed plants (Figure 1). These beneficial effects of Si on the biomass of S-deprived plants agree with earlier studies showing that Si alleviates the negative effects on growth of plants experiencing macronutrient deficiencies (for review see [27]) such as N [26], K [38] or P [39,40]. However, with the beneficial effects of Si being poorly documented in plants subjected to S-deficiency [27,37], this study is the first to demonstrate enhancement of growth of S-deprived clover plants following application of Si. In addition, in our study, it was surprising to find that Si supply also promoted the growth of S-fed plants by enhancing shoot biomass from D42 to D63 (Figure 1). Indeed, in the literature it is usually stated that the benefits of Si for growth are mainly observed in plants under biotic and abiotic stresses [41].

To explain these beneficial effects of Si in *Trifolium incarnatum*, our study has focused on N nutrition, which is one of the major macronutrients and considered the most important component for supporting plant growth [42,43]. Our results show that at D63, compared to S-fed plants, the S-deprivation in -S-Si plants resulted in a significant decrease in the total amount of N, and this was mainly explained by a significant decrease in N_2_ atmospheric fixation while the uptake of N from the nutrient solution was not affected (Figure 4). This decrease in N_2_ fixation is associated with a decrease in the number of nodules, the biomass of individual nodules and the total biomass of nodules per plant (Figure 3). Further, it was also associated with a significant decrease in total nitrogenase abundance (Figure 5) in roots in -S-Si plants, which occurred after 21 days of S-deficiency. These negative effects of S-deficiency on N_2_ fixation, the number and biomass of nodules the nitrogenase abundance all agree with previous studies [44,45]. In particular, there are close parallels with work performed on *Trifolium repens* by [8], who reported that S-deficiency reduced cysteine and methionine contents, both of which are key amino acids for nitrogenase synthesis. As previously reported, these responses to S-deprivation could be explained by an increase in ROS [46], which will directly impact the structures and the activity of nodules [47]. Compared to -S-Si plants, supplying Si to S-deprived plants (-S+Si) led to an increase in the amount of total N, mainly due to the strong increase in N_2_ fixation from D42 onwards (Figure 4B). This improvement in N_2_ fixation was associated with an increase in number and total biomass of nodules from D21 (Figure 3) and later (from D42) with a higher nitrogenase abundance in the nodules (Figure 5 and Appendix A) of -S+Si plants relative to -S-Si plants. This beneficial effect of Si agrees with numerous studies that have reported exogenous Si supply as being one of the most effective strategies to enhance nodulation and N_2_ fixation in legumes under stressed conditions, including metal toxicity, salinity, alkalinity and pathogen attack [29,30,48]. Since nitrogen is a macronutrient required for plant growth, the increase in nitrogen fixation caused by Si leads to an increase in the amount of total nitrogen -S+Si plants (Figure 4) and improves their growth (Figure 1). However, our study is currently the only one to show these beneficial effects of Si in response to S-deficiency. Given the central role of S in numerous biological processes (such as N_2_ fixation, nitrogenase and leghemoglobin synthesis and ROS detoxification), it is therefore difficult to explain how a Si supply is able to alleviate the negative impacts of S-deprivation, and particularly regarding nodulation and N_2_ fixation. In addition, in S-fed plants, supplying Si also led to an enhancement of the total N and N_2_ fixation (Figure 4) without any effect on the number and biomass of nodules (Figure 3). In this case, the improvement in N_2_ fixation is essentially due to the strong increase in the nitrogenase concentration in nodules (S2). This result agrees with a previous study showing that a Si application on *Medicago truncatula* increased the nitrogenase concentration in nodules and improved their N_2_ fixation capacity [30]. 

The current study has shown clearly that supplying Si alleviates the negative effects of S-deprivation in *Trifolium incarnatum*, especially by promoting root nodulation and N_2_ fixation. Nevertheless, it is very difficult to explain the mechanisms related to these Si beneficial effects without being speculative. Indeed, regardless of the explanations advanced by earlier studies on other abiotic stresses where beneficial effects of Si on nodulation and N_2_ fixation in legumes have been reported, the underlying mechanisms remain unclear and only some assumptions can be formulated [29]. For example, [30] reported that Si supplementation might affect key symbiotic signals such as flavonoid compounds and would promote the attraction of rhizobia and increase the degree of nodulation. Another assumption supported by the same authors concerned the silicification of nodules, which would enhance their N_2_ fixation efficiency. This hypothesis is compatible with the preferential accumulation of Si in the roots of *Trifolium incarnatum* (Figure 2C) observed in this study. Finally, it is well known that S-deprivation leads to ROS production, which might be significantly reduced by Si supply as shown recently by [28]. As suggested by some authors, the decrease in oxidative stress in -S+Si plants can lead to better nitrogenase activity, with this enzyme known to be inhibited in the presence of ROS [30,35]. Even if our pioneering study has demonstrated for the first time that Si supply alleviates the negative effect of S-deficiency in *Trifolium incarnatum*, further investigation is required to verify all or part of these hypotheses and to decipher the fine mechanisms that remain unclear or misunderstood.

In conclusion, the results of this study suggest that Si fertilization could be considered as part of emerging agroecological and agricultural practices to optimize fertilizer inputs. For example, to reduce the quantity of N inputs applied during *Brassica napus* crop growth, several recent studies have explored mixed cropping of this plant with legume species [49], and in particular with *Trifolium incarnatum* [18,33]. Indeed, the high sulfur requirements of *Brassica napus* for growth can decrease the availability of S in the soil and lead to decreases in the N_2_ fixation capacity of the companion legume. Thus, as suggested by this study, Si fertilization could alleviate this by increasing the atmospheric N_2_ fixation capacity of clover and eventually lead to an enhancement of the agronomic performance of this mixed cropping by improving the transfer of exuded N compounds from *Trifolium incarnatum* to *Brassica napus* [34]. 

## 4. Materials and Methods 

### 4.1. Plant Growth Conditions and Experimental Design

The experimental design is summarized in Figure 6. In a greenhouse, seeds of *Trifolium incarnatum* L. were germinated on perlite over deionized water for four days in the dark [18]. When the first leaf emerged, seedlings were transferred to natural light conditions and supplied with nutrient solution for two weeks containing: KNO_3_ (1 mM), KH_2_PO_4_ (0.25 mM), KCl (1 mM), CaCl_2_ (3 mM), MgSO_4_ (0.5 mM), EDTA-2NaFe (0.2 mM), H_3_BO_3_ (14 µM), MnSO_4_ (5 µM), ZnSO_4_ (3 µM), CuSO_4_ (0.7 µM), (NH_4_)_6_Mo_7_O_24_ (0.7 µM), CoCl_2_ (0.1 µM). After two weeks, the seedlings were transplanted in hydroponics tanks (20 L, nine plants per tank) containing the nutrient solution described above and inoculated with *Rhizobium leguminosarum bv trifolii* (T354) [33] previously grown in YEM liquid medium [50] for 72 h. After four weeks (D0), plants were separated into four sets: the first two sets corresponded to plants maintained on S (+S; 500 µM) and supplied (+S+Si) or not (+S-Si) with 1.7 mM silicon for 63 days (D63). The nutrient solution described above with the addition of 1.7 mM silicon (Si as sodium metasilicate: Na_2_SiO_3_) or NaCl (3.4 mM to compensate the sodium supplied by the Si treatment) to the +S+Si and +S-Si plants, respectively. The other two sets corresponded to plants deficient in S (-S) and supplied (-S+Si) or not (-S-Si) with silicon for 63 days. For this, the nutrient solution was modified as follows to remove S: KH_2_PO_4_ (0.25 mM), KCl (1 mM), CaCl_2_ (3 mM), MgCl_2_ (0.5 mM), EDTA-2NaFe (0.2 mM), H_3_BO_3_ (14 µM), MnCl_2_ (5 µM), ZnCl_2_ (3 µM), CuCl_2_ (0.7 µM), (NH_4_)_6_Mo_7_O_24_ (0.7 µM) and CoCl_2_ (0.1 µM). In addition, plants were supplied with K^15^NO_3_ (1 mM, 0.5% isotopic excess) in order to discriminate between the N taken up from nutrient solution and the N derived from atmospheric fixation in the plants. Throughout the culture duration, natural light was supplied by high pressure sodium lamps (Philips, MASTER GreenPower T400W) with photosynthetically active radiation of 450 μmol photons·m^−2^·s^−1^ at canopy height. Nutrient solutions, continuously aerated with a compressed air bubbling system, were replaced every three days. The pH of the nutrient solution was monitored each day and adjusted if necessary to 5.8 ± 0.2. At each solution renewal, 10 mL of medium containing *Rhizobium leguminosarum* bv *trifolii* (T354) was added to each tank. After 21, 42 and 63 days of cultivation, plants were harvested (Figure 6). At each harvest, the nodules were counted, separated from roots, weighed and frozen at −80%. In addition, shoots and roots were separated and weighed. Then, an aliquot of each plant compartment was placed in liquid nitrogen and stored at −80 °C and the rest was dried at 60 °C. After dry weight determination, samples were ground to perform the elemental analyses. 

### 4.2. Determination of Total, S, Si, N and ^15^N Concentrations and Calculation of the Distribution of Nitrogen Sources in Plants

For S and Si concentration determinations, approximately 1 g of the dried powder was analyzed with an X-ray-fluorescence spectrometer (XEPOS, Ametek, Berwyn, PA, USA) using calibration curves obtained from international standards.

To determine the total N and ^15^N concentrations, 1.5 mg of each dried powder was precisely weighed and placed into tin capsules before analysis with a continuous flow isotope ratio mass spectrometer (IRMS, Horizon, NU Instruments, Wrexham, UK) linked to a C/N/S analyzer (EA3000, Euro Vector, Milan, Italy). The total N amount (*Ntot*) in each organ or in the whole plant was calculated as: $$Ntot=\frac{\%N\;x\;W}{100}$$

The amount of ^15^N (*^15^N*) was determined by the following:$${}^{15}N=\frac{{\%}^{15}N\;x\;Ntot}{100}$$

As the nutrient solution was labeled with a 0.5% isotopic excess, it was possible to calculate the total amount of nitrogen taken up from the nutrient solution (*N_ns_*):$$N_{ns}=\frac{{}^{15}\mathrm{Ntotal}\;x\;100}{0.5}$$

The amount of nitrogen from dinitrogen (N_2_) atmospheric fixation (N_atm_) was then obtained by subtracting the amount of nitrogen taken up from the nutrient solution from the total nitrogen in the plant. Then, the proportion of each N source (% of N_ns_ or % N_atm_) was calculated as follows:$${\%N}_{ns}(\mathrm{or}\;{\%N}_{atm})=\frac{N_{ns}\left(\mathrm{or}\;N_{atm}\right)\;x\;100}{N_{tot\;(\mathrm{whole}\;\mathrm{plant})}}$$

### 4.3. Extraction and Quantification of Proteins from Nodules

Total proteins were extracted from nodules using the protocol described by [8]. Fresh nodules were ground in liquid nitrogen in a mortar and approximately 30 mg of the resulting powder was mixed in 2 mL of cold acetone containing 10% trichloroacetic acid (*w*/*v*). This mixture was centrifuged at 16,000× *g* (4 min at 4 °C), the supernatant was then removed the pellet was resuspended with 1.75 mL of ammonium acetate (0.1 M)/methanol (80%) buffer to precipitate the proteins. After centrifugation at 16,000× *g* (3 min at 4 °C), the supernatant was removed, and the pellet was washed with 1 mL of acetone (80%) and centrifuged again. The resulting pellet was resuspended with 800 µL of phenol pH 8 and 800 µL of sodium dodecyl sulfate (SDS) buffer [30% saccharose (*w*/*v*), 2% SDS (*w*/*v*), 0.1 M TRIS-HCl (*w*/*v*), 0.5% β-mercaptoethanol (*v*/*v*), pH 8] then centrifuged at 16,000× *g* (10 min at 4 °C). The upper phenolic phase was recovered and 800 µL of ammonium acetate (0.1 M)/methanol (80%) buffer was added. After storing overnight at −20 °C, the mixture was centrifuged at 16,000× *g* (10 min at 4 °C). Then the pellet was washed with 1 mL of methanol (100%) and after centrifugation at 16,000× *g* (10 min at 4 °C) it was washed again with 1 mL of acetone (80%). The pellet was resuspended with a buffer containing dithiothreitol (DTT, 11.11 mM), thiourea (2.22 M), urea (6.66 M) and Tris buffer 2 M (33.33 mM) and used for the determination of the protein concentration using the Bradford method [51].

### 4.4. Immunodetection and Quantification of Nitrogenase in Root Nodules

The abundance of nitrogenase in nodules was determined after Western blotting of 20 µg of total proteins previously separated by SDS-PAGE (12% polyacrylamide gels at a constant current: 250 V, 75 mA, 1 h). Western blots were carried out on a polyvinylidene fluoride (PVDF) membrane (Immobilon-PVDF, Millipore Bedford, USA) by semi-dry electroblotting (Pierce™ G2 Fast Blotter, Thermo Fisher Scientific, Waltham, USA). Conditions for protein transfer and immunoblotting were carried out as previously described [8]. Briefly, after Western blotting, PVDF membrane was incubated overnight with chicken polyclonal anti-nitrogenase (NifH) from IgG antibodies (Agrisera, SE 911, Vanadas, Sweden; dilution 1:2000). Subsequently the PVDF membrane was washed four times with TBST (Tris-base 10 mM, NaCl 150 mM, pH 8, 0.1% Tween 20) before incubation with secondary antibody (rabbit anti chicken immunoglobin Y coupled to phosphatase alkaline; dilution 1:6000) for 1 h. The nitrogenase–antibody complex was revealed using the Bio-Rad alkaline phosphatase kit as a single signal at a molecular weight between 25 and 37 kDa (S2). Independently performed gels and Western blots were scanned and analyzed using the Millipore Bioimage computerized image analysis system to determine the intensity of signal (U.A.) that was proportional to the nitrogenase abundance in each nodule sample. The nodule nitrogenase concentration (U.A. g^−1^ FW) was calculated by dividing the intensity value by the fresh nodule biomass (20 µg) used for Western blot analysis. Then the total abundance of nitrogenase in all the nodules of one plant was estimated by multiplying the nitrogenase concentration by total nodule biomass of this plant. Finally, at each harvest time, the results were expressed relative to the abundance of nitrogenase in the total nodules from control plants (+S-Si).

### 4.5. Statistical Analysis

The experiment was performed with five independent replicates. All data are indicated as the mean ± S.E (*n* = 5). Statistical analyses were performed using R software (version 4.2.0: R Core Team, 2022). At each harvest date (D21, D42 and D63), significant differences between treatments were determined using Student’s *t*-test (*p* < 0.05).

## Figures and Tables

**Figure 1 plants-12-02248-f001:**
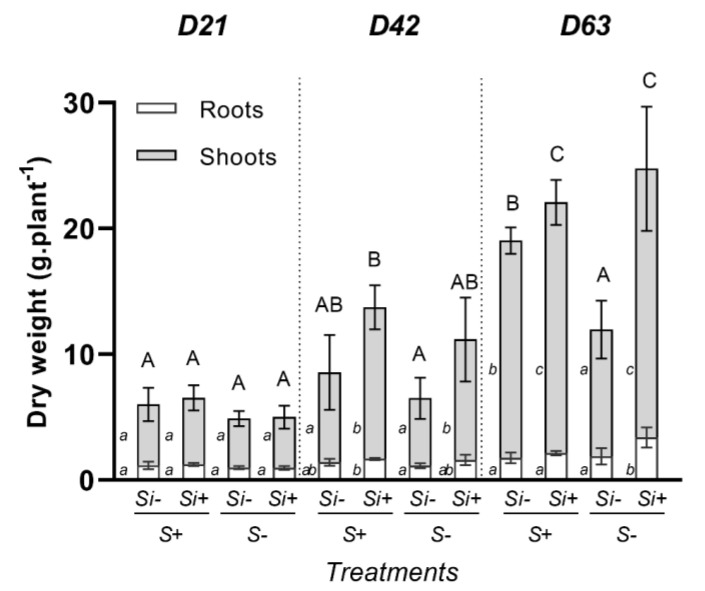
Root and shoot dry weights of *Trifolium incarnatum* plants cultivated in hydroponic conditions for 21 (D21), 42 (D42) and 63 days (D63) with (+S; 500 µM) or without (-S) sulfur and supplied with (+Si; 1.7 mM) or without (-Si) Si. Data are means ± SE (*n* = 5). Different lowercase and uppercase letters indicate that the mean dry weight between organs or whole plants harvested at the same time (D21, D42 or D63) are significantly different (*p* < 0.05), respectively.

**Figure 2 plants-12-02248-f002:**
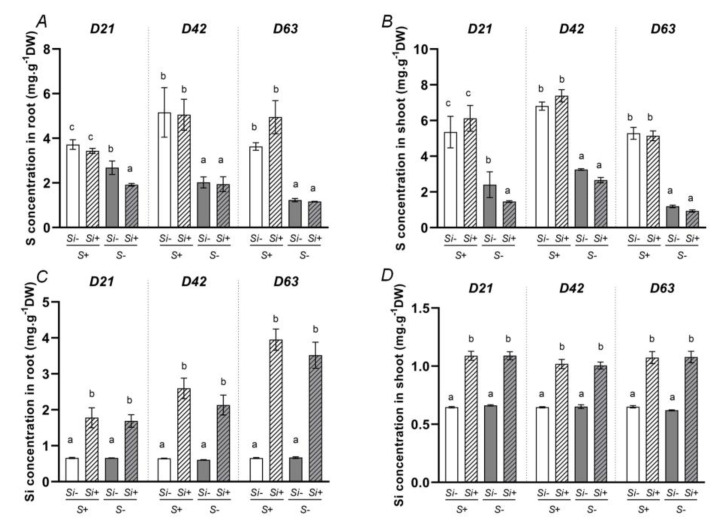
Sulfur (**A**,**B**) and silicon (**C**,**D**) concentrations in roots and shoots of *Trifolium incarnatum* plants cultivated in hydroponic conditions for 21 (D21), 42 (D42) and 63 days (D63) with (+S; 500 µM) or without (-S) sulfur and supplied with (+Si; 1.7 mM) or without (-Si) Si. Data are means ± SE (*n* = 5). Different lowercase letters indicate that the means between plants harvested at the same time (D21, D42 or D63) are significantly different (*p* < 0.05).

**Figure 3 plants-12-02248-f003:**
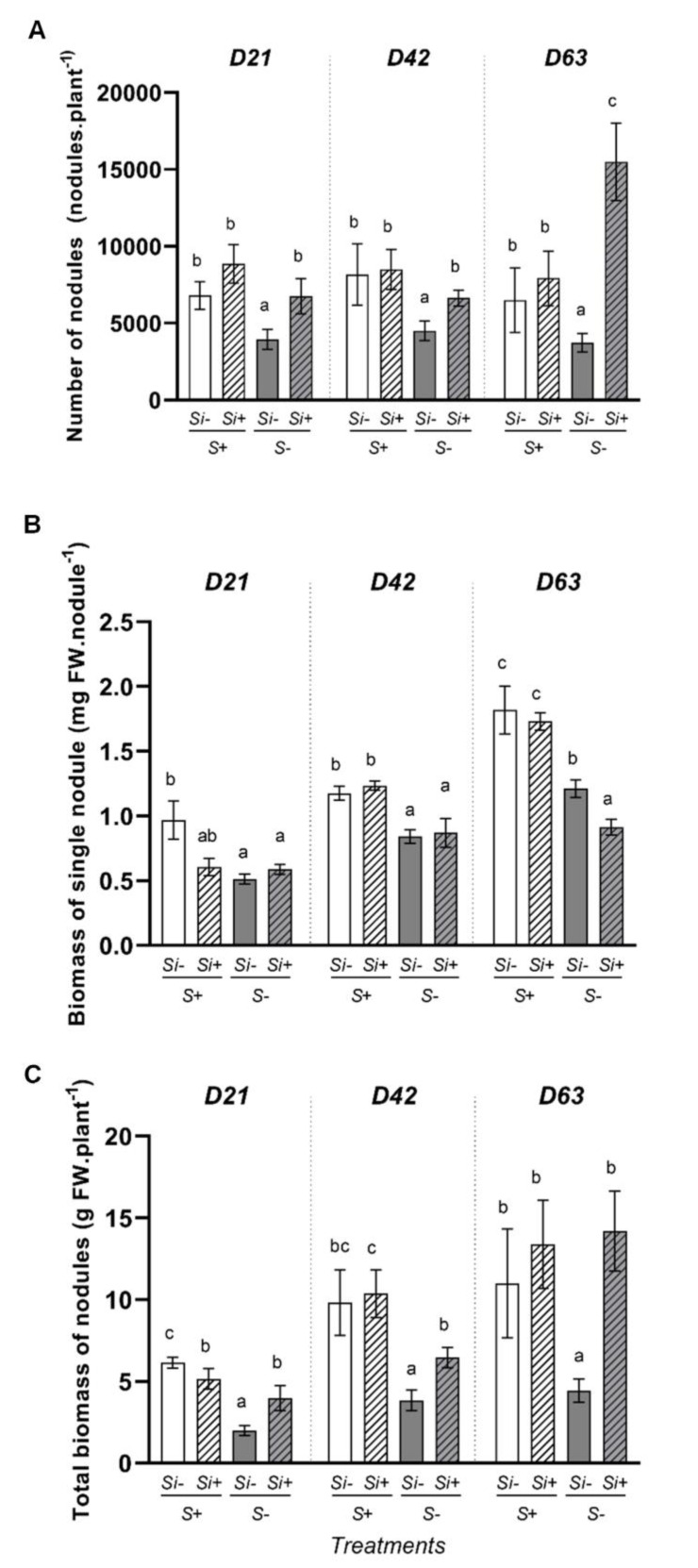
Number of nodules (**A**), fresh weight of individual nodules (**B**) and total fresh weight of all the nodules (**C**) of *Trifolium incarnatum* plants cultivated in hydroponic conditions for 21 (D21), 42 (D42) and 63 days (D63) with (+S; 500 µM) or without (-S) sulfur and with (+Si; 1.7 mM) or without (-Si) Si supply. Data are means ± SE (*n* = 5). Different lowercase letters indicate that the means between plants harvested at the same time (D21, D42 or D63) are significantly different (*p* < 0.05).

**Figure 4 plants-12-02248-f004:**
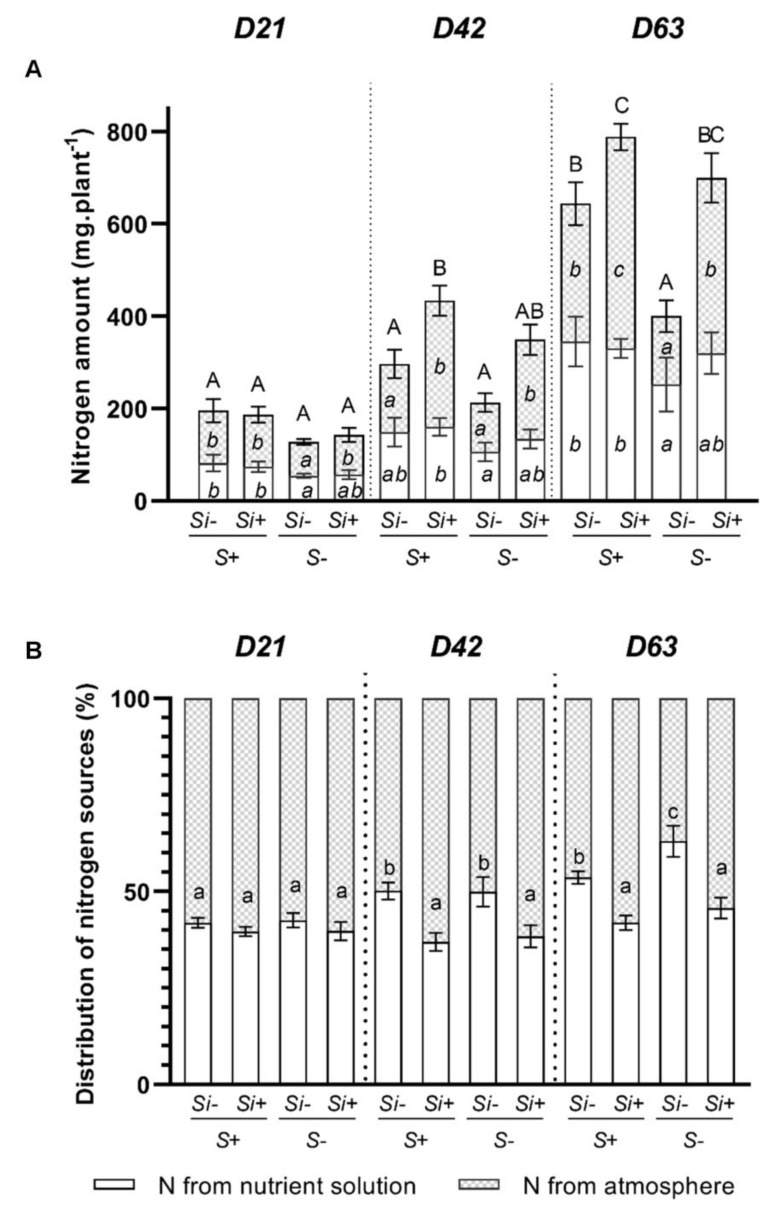
Root and shoot total nitrogen amounts (**A**) and distribution of nitrogen sources (**B**) in *Trifolium incarnatum* plants cultivated in hydroponic conditions for 21 (D21), 42 (D42) and 63 days (D63) with (+S; 500 µM) or without (-S) sulfur and with (+Si; 1.7 mM) or without (-Si) Si. Data are means ± SE (*n* = 5). Different lowercase italic letters to the left of the histogram and different uppercase letters above or inside of the histogram indicate that the means between plants harvested at the same time (D21, D42 or D63) are significantly different (*p* < 0.05).

**Figure 5 plants-12-02248-f005:**
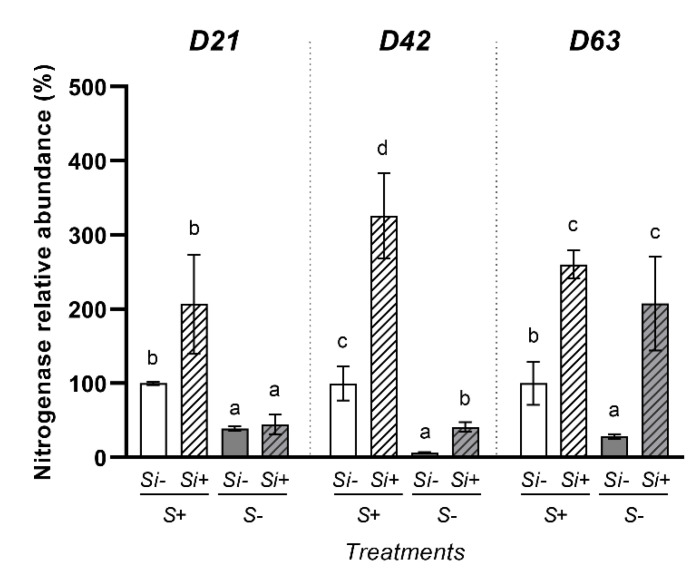
Change in the total amount of nitrogenase in nodules of *Trifolium incarnatum* cultivated in hydroponic conditions for 21 (D21), 42 (D42) and 63 days (D63) with (+S; 500 µM) or without (–S) sulfur and with (+Si; 1.7 mM) or without (-Si) Si supply. For each time (D21, D42, or D63) and each treatment the total amount of nitrogenase was expressed relative to the value obtained for the +S-Si plants. Data are means ± SE (*n* = 3). Different lowercase letters indicate that the mean between plants harvested at the same time (D21, D42 or D63) are significantly different (*p* < 0.05).

**Figure 6 plants-12-02248-f006:**
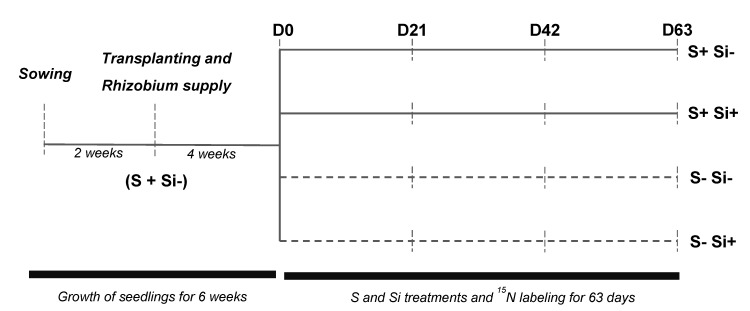
Experimental design used to study the effect of silicon supply on *Trifolium incarnatum* L. grown with (+S) or without (-S) sulfur. After six weeks of cultivation with sulfur (+S; 500 µM) (D0), plants were separated into four batches: the first two batches (full line) corresponded to plants that continued to be fed with S (+S; 500 µM) and supplied (+Si; 1.7 mM) or not (-Si) with Si for 63 days (D63). The other two batches (dotted lines) corresponded to plants deficient in S (-S) and supplied (+Si; 1.7 mM) or not (-Si) with silicon. At D0, all plants were fed with K^15^NO_3_ (1 mM, 0.5% isotopic excess).

## Data Availability

Not applicable.

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
