# Peer review of "Silicon Supply Improves Nodulation and Dinitrogen Fixation and Promotes Growth in Trifolium incarnatum Subjected to a Long-Term Sulfur Deprivation"

_plants, 2023, doi:10.3390/plants12122248_

Round 1

Reviewer 1 Report

The manuscript of Coquerel et al. investigates the effect of Si supply on root nodulation and N2 fixation capacity in Trifolium subjected to S deficiency. This study is novel and generally well conducted, and the presented data are sound. However, the mechanisms of how Si alleviated Si deficiency and recovered root N2 fixation machinery remained still unclear. Despite lacking in mechanistic explanation, my first impression of the study is excellent, however after careful reading, one could find some methodological aspects that need to be clarified: 

1. Did you check the real concentration of mono-silicic acid in the nutrient solution? Did you monitor the pH of the nutrient solution and how frequently? 

2. Is there any recovery of the S concentration determined by CNS and XRF? The reason why you chose XRF for S determination should be explained.  

Minor points:

1. What you presented in Figure 2 is concentration rather than content. So, change it in the figure and throughout the text accordingly. 

2. Please consolidate y-legends (-S/+S versus +/- S) of SD1 with SD2 and other figures. 

Please re-check grammar and typos.

Author Response

Dear reviewer,

Best regards

Reviewer 2 Report

Manuscript entitled "Silicon supply improves nodulation and dinitrogen fixation and promotes growth in Trifolium incarnatum subjected to a long-term sulfur deprivation" investigated the effect of silicon supply in Trifolium incarnatum subjected (or not) to a long-term S deficiency (63 days). Overall the topic is interesting, but there is some points:

Abstract:

This section has written well.

Introduction:

This section need to expand, the authors should compare the importance of their research with previous researches and bold the lack of previous researches.

Materials and methods:

This section should be written before the results.

Lines292-293: provide reference(s)

Lines: 317-318: why did you choose 21, 42, 63 days for the growth time? Provide reference (s)

Results and discussion:

These sections have written well.

Conclusion:

Please add the separate conclusion section for your reserach (after discussion).

Minor editing of English language required.

Author Response

Dear reviewer,

Best regards
